# Influence of the Printing Orientation on Parallelism, Distance, and Wall Thickness of Adjacent Cylinders of 3D-Printed Surgical Guides

Aisha Ali [1], Hossein Bassir [2] and Rafael Delgado-Ruiz [1,*]

1   School of Dental Medicine, Stony Brook University, Stony Brook, NY 11794, USA
2   Private Practice in Periodontics, 2319 Rayford Rd Ste 100, Spring, TX 77386, USA
*   Correspondence: rafael.delgado-ruiz@stonybrookmedicine.edu; Tel.: +1-6316326913

**Abstract:** This in-vitro study aimed to evaluate the influence of the printing orientation on parallelism, distance, and thickness between adjacent cylinders of 3D-printed surgical guides. CAD software was used to design a surgical guide with two adjacent parallel cylinders (reference); the design was saved as standard tessellation software (STL) and 63 samples were printed using three different orientations (0, 45, and 90 degrees). A metrology digital microscope was used to measure the distance, the angle and the thickness of the guides cylinders. Afterwards, the printed guides were scanned and cloud comparison software was used to compare STL files from the printed guides against the reference CAD model. One-way analysis of variance and Tukey test were used for multiple comparisons between groups and significance was $p < 0.05$. The printing orientation affected the distance between cylinders, the parallelism and the wall thickness. In addition, there were global deviations in all printing orientations. Printing with 90 degrees orientation produced almost-parallel cylinders but walls thicker than the reference model; all the cylinders converged toward the coronal but printing at 0 degrees produced the closest distance to the reference value. Within the limitations of this experimental in-vitro study it can be concluded that all the printing orientations influence the angle, the distance, and the thickness between adjacent cylinders of a surgical guide. Printing at 90 degrees produces the best global correspondence with the master model.

**Keywords:** 3D-printing; accuracy; guide cylinders; surgical guide; errors; dentistry; dental implants; implantology

## 1. Introduction

Malposition of dental implants is one of the major causes for future prosthetic, and peri-implant hard and soft tissue problems [1]. Improper placement of dental implants can not only result in sub-optimal esthetic outcomes, but also can affect their cleanability, which can result in inflammation of the peri-implant tissues [2]. Several factors such as limited mouth opening, asymmetric bone contours, or clinician's lack of experience may cause deviations in implant positioning [3]. Methods to achieve a more precise implant placement were implemented, and surgical templates became the standard of care [4]. Initially, these templates were fabricated using vacuum forming methods or using acrylic resins based in polymethylmethacrylate (PMMA), and their use demonstrated reduced surgical and prosthetic complications [5].

Computer-aided design and manufacturing (CAD-CAM) surgical templates have been implemented because they provide more precise dental implant placement and minimize positional errors compared to previous surgical guides [6], thus leading to more predictable restorations [7]. Within CAM, methods for the fabrication, milling, and printing offer repeatability and precision [8]. Moreover, guided implant surgery compared to conventional methods of implant surgery have demonstrated fewer deviations, and in general a better positioning of the implants [9–11].

However, there are certain cumulative errors in guided implant surgery that contribute to the deviation between the initial digitally planned position and the final implant location [12,13]. Some of the sources of error are: inadequate data acquisition, failures during the image process, improper scanning, inaccurate file merging during digital planning, errors during the fabrication of surgical guides, and incorrect implant surgery [14–19].

Specifically, additive manufacturing (commonly referred to as 3D printing) provides smaller and more economical desktop printers in comparison to subtractive manufacturing devices [20]. Two methods with generalized use for 3D-printing surgical guides based on Polymethyl Methacrylate resins (PMMA) are stereolithography (SLA), and digital light processing (DLP) [21]. The SLA printer use an ultraviolet (UV) laser to trace the outline of the object layer by layer [22]. The DLP printer applies a UV light pattern from a digital projector, creating a single image for each layer, and thus is faster than SLA printers [23]. Once the digital design of the object is completed, the printing orientation is determined together with the layer thickness and the supporting structures [24–27].

Factors that have been reported to affect the precision of 3-D printed surgical guides include: guide design, parameters of fabrication such as light intensity and exposure time, and printing orientation [28,29]. The printing orientation affects the way the object is layered and the number of layers to be printed [30]. In addition, the number of models placed on a build platform depends on the shape, size, and printing orientation [31]. Previous studies have evaluated the effects of various printing angulations on the accuracy of surgical guides with variable results favoring printing orientations of 45° or 90° [32,33]. The body of a surgical guide possesses certain common elements: an intaglio that contacts the supporting structures (teeth, mucosa, or bone), a cameo surface (exposed to the operator and oral environment), windows (to evaluate the seating of the guide), and two types of cylinders [34]. The first type (fixation) serves to hold the bone anchoring pins, and the second type (guiding cylinders) holds the metallic sleeves that guide the implant drills and implant insertion [35].

However, parameters like arch curvature, type of guide support, thickness and extension of the guide, and the number of implants and cylinders are different between patients. This makes it difficult to determine the specific impact of isolated parameters on the cumulative errors in guide implant surgery. To the best of the authors' knowledge, no studies have evaluated the isolated effect of the printing directions (0°, 45°, 90°) on parallelism, distance, and wall thickness of adjacent cylinders of 3D-printed surgical guides.

Therefore, the present in vitro study aimed to study the influence of printing orientation on the parallelism, distance, and the thickness between adjacent cylinders of a surgical guide. The null hypothesis was that the printing directions (0°, 45°, 90°) have no influence on the orientation of adjacent cylinders of surgical guides.

## 2. Materials and Methods

In this in-vitro study, three experimental groups based on the printing orientation were created: (a) 0 degrees, (b) 45 degrees, and (c) 90 degrees. The sample size was determined with the software Raosoft (Raosoft, Inc., Seattle, WA, USA) at the website http://www.raosoft.com/samplesize.html, accessed on 7 July 2022. The following data were inserted in the online formulary, a 5% margin of error, a 95% confidence level, and a standard deviation (SD) of 0.5. The sample size was determined as N = 63. Each experimental sub-group group was set for 21 samples.

Fusion 360 (version October 2022, Autodesk Inc., San Rafael, CA, USA) and AutoCAD 2022 (version 24.1, Autodesk Inc., San Rafael, CA, USA) software were used to design a model simulating a surgical guide with two cylinders for two adjacent implants with the following characteristics: rectangular base (3 mm thickness × 13 mm width × 30 mm length); two adjacent cylinders with an inner diameter of 8 mm, an outer diameter of 9.2 mm and a height of 10 mm. The distance between the walls of two adjacent cylinders was 4 mm. The distance between the center of the cylinder and the width of the rectangular

prism was 8.4 mm. The distance between the center of the cylinder and the length of the rectangular prism was 6.5 mm (Figure 1).

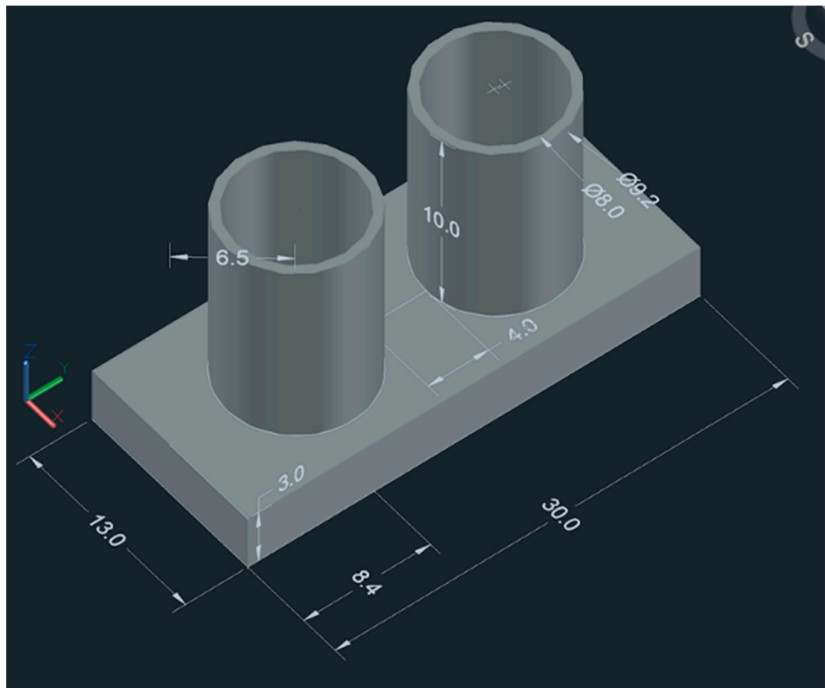

**Figure 1.** Scheme of the CAD design (dimensions in mm).

The design was exported as a master STL file and transferred to the printing preparation software (Preform Software, version 3.0.1; Formlabs, Somerville, MA, USA); the design was replicated 21 times per each printing angle (0, 45, and 90 degrees) (Figure 2).

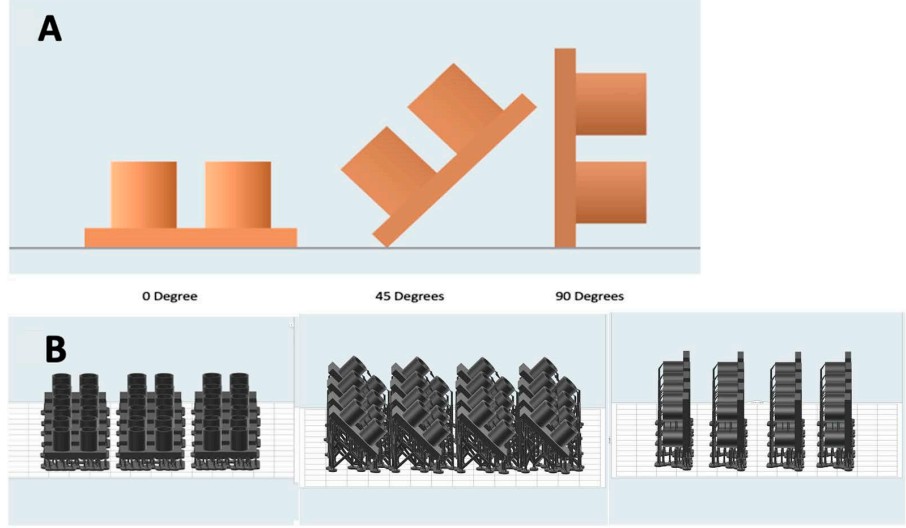

**Figure 2.** (**A**) Printing orientations of the surgical guides selected for this experiment. (**B**) Surgical guides with their supports respective to the build platform.

Models were arranged so that all samples for a single printing orientation could be printed in one cycle. The support type used was mini rafts with internal supports at a density of 0.50 and a touch size of 0.40 mm. The models were printed using an SLA Form2 3D printer (Formlabs, Somerville, MA, USA) and photopolymerization resin (Dental SG Resin, RS-F2-SGAM-01; Formlabs, Somerville, MA, USA) at a layer thickness of 100 μm.

Post-fabrication, the samples were washed following the manufacturers' instructions in 99% isopropyl alcohol for 20 min (Form Wash; Formlabs, Somerville, MA, USA). After washing, the samples were removed from the build platform and transferred to a curing unit (Form Cure; Formlabs, Somerville, MA, USA) and post-cured for 1 h at 50 °C. Flush cutters were used to remove the support structures from each sample. All samples were subsequently placed and stored away from ambient light. If a sample was broken during post processing, then the sample was reprinted and cleaned. The primary endpoint of this experimental study was to evaluate the dimensional properties of the cylinders of simulated surgical guides printed with different orientations. The outcomes included measuring the angle and distance between cylinders (expressed in mm), the cylinders' wall thickness (mm), and the global positive and negative deviations (root mean squared error) of the whole printed samples. Two methods used to evaluate the accuracy of the samples were a direct measurement using a digital microscope (Keyence VHX-6000; Keyence, Itasca, MN, USA) and a global evaluation using dimensional inspection software (GeoMagic Design X; 3D Systems, Rock Hill, SC, USA) (version. 2020.0.4).

## 2.1. Direct Measurement

The digital microscope was used at a magnification of $20\times$ to measure the following characteristics: the angle between cylinders (parallelism), the distance between the cylinders, and the thickness of the cylinder walls. The samples were oriented flat to the surface and with the top of the cylinders facing up. The angle between cylinders was measured by tracing lines parallel to the adjacent cylinder walls. The angle formed between both lines was automatically recorded by the microscope (Figure 3).

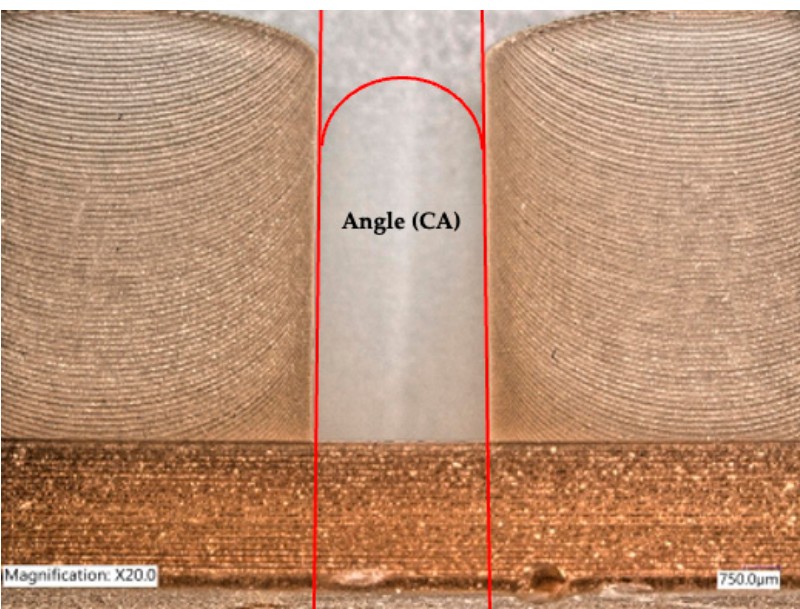

**Figure 3.** Angle between adjacent cylinders.

The distance between cylinders was evaluated from a lateral view. Lines were traced at the adjacent walls of the cylinders and the distances at the coronal and at the base between the walls of the cylinders were recorded. In addition, the distance between the cylinder walls at the top (DC) was evaluated by tracing a line perpendicular to the centers of the cylinders; the site where the line intersects the edge of each cylinder was marked and the distance between the cylinder walls was measured (Figure 4).

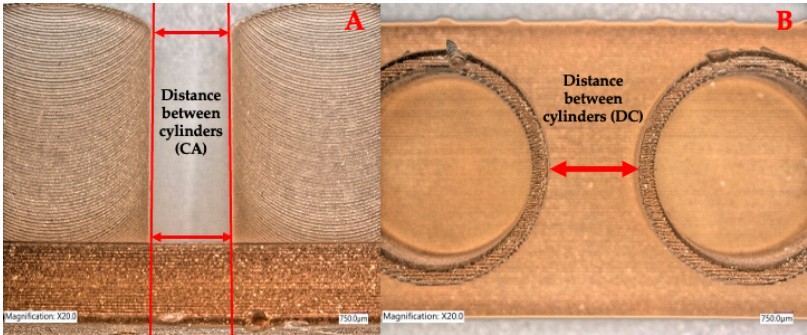

**Figure 4.** Distance between cylinders. (**A**) represents a lateral view of the distance between the cylinders' walls at the top and at the base. (**B**) represents a top view at the top of the cylinders.

The thickness of the cylinder walls was measured by tracing two lines that divided each cylinder in four segments. The wall thickness was measured at the four points where the lines intersect the cylinder walls. The measures were obtained in microns (Figure 5).

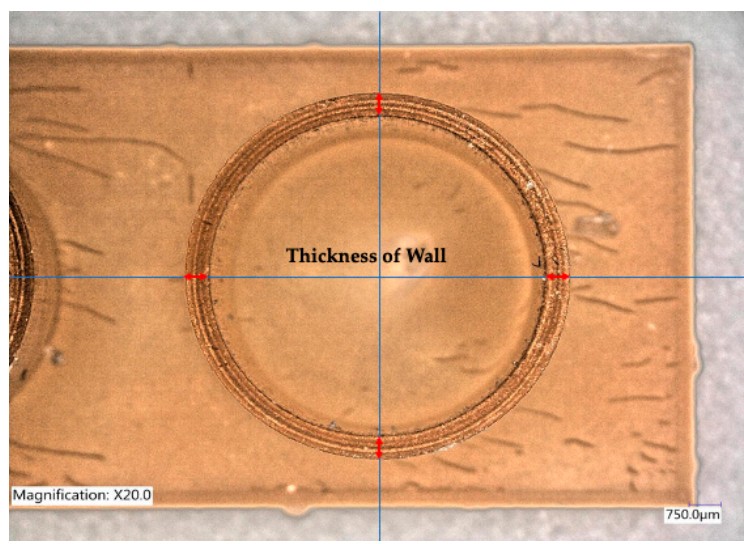

**Figure 5.** Thickness of the cylinders' walls measured at four equidistant points.

## 2.2. Global Evaluations

For the evaluation of the global dimensions, the printed samples were covered with a thin, non-reflective layer of zirconium powder prior to scanning. Then each sample was placed in a holder supporting 3 mm of the rectangular base and inserted into a laboratory scanner (E3, 3Shape, Copenhagen, Denmark). The scanning was completed in detail mode; afterward, the scanned files were digitally trimmed using the 3Shape TRIOS Design Studio (E3, 3Shape, Copenhagen, Denmark) to remove the contours of the sample holder. Cloud comparisons between original master design and each scanned guide were completed by superimposing the master STL file to the scanned STL files.

The global evaluations were completed by using surface-matching (cloud comparison) software (GeoMagic Design X; 3D Systems, Rock Hill, SC, USA) (version 2020.0.4). The master STL file was imported and moved to reference data. The resegment, split, and merge tools were used to segment the reference model into 5 separate regions: the base, left hollow cylinder, right hollow cylinder, base inside left cylinder, and base inside right cylinder (Figure 6).

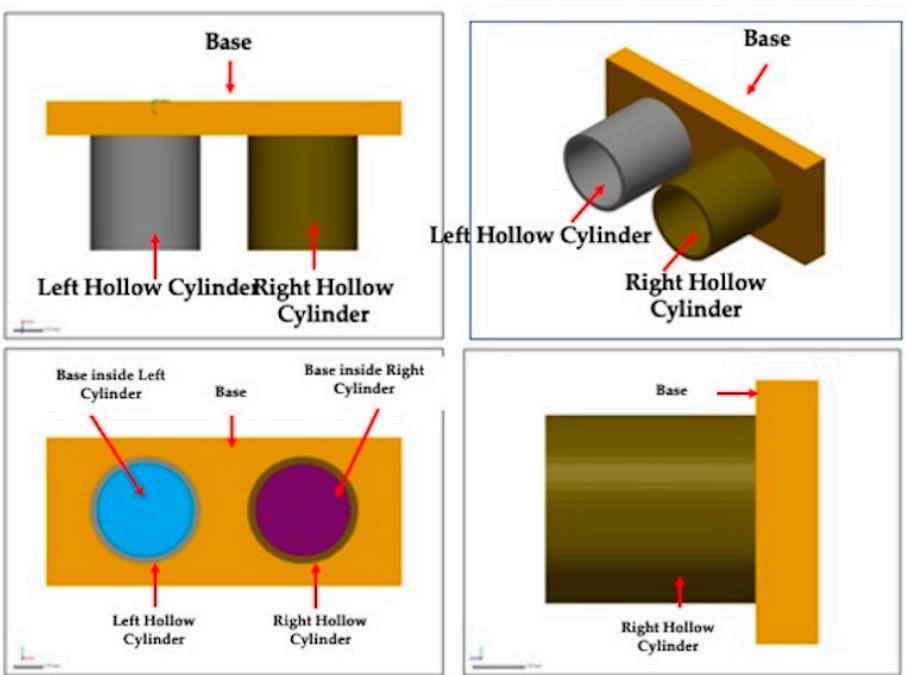

**Figure 6.** Regions identified at the reference model for cloud comparisons with the STL files of the 3D-printed samples.

The STL files from the printed samples were imported one by one. Each sample was first aligned using the transform alignment; then, the N points method was used to superimpose the two files. The reference and the superimposed STL files were indicated in two windows where 10 points were placed on each model for alignment. Five points were placed at the top of each cylinder and 5 points were placed using the crosshairs of the cursor at each corner and/or middle of the base. The 3D comparison was conducted using the shape method with a 100% sampling ratio, shortest projection direction, and maximum deviation of 1 mm. Reports generated for each superposition included the root mean square error (RMSE), mean negative, and mean positive deviations.

Given that printing at different angles may affect angulation, parallelism (cylinders may diverge or converge in a coronal and apical direction) and thickness, positive and negative global deviations were considered. The positive average deviations indicate the mean of all the positive gap distances between superimposed clouds of points, and the negative average deviations indicate the mean of all the negative gap distances between superimposed clouds of points. The root means square error values were measured to determine the magnitude of all deviation values at the cylinders and at the base.

Statistical analyses were performed using the web application Minitab. The normality of the samples was evaluated with the Kolmogorov-Smirnoff test. One-way analysis of variance (ANOVA) was completed, and differences between groups were evaluated with Tukey post-test. T-test was used to determine differences between the global, positive, and negative deviations. The level of significance for all analyses was set to $p < 0.05$.

### 3. Results

Printing at different orientations affected the angulation between cylinders, distance between cylinders, and thickness of cylinders of 3D printed surgical guides as confirmed with direct digital measurements and cloud comparisons of the mesh structures. In the next subsections the results for each variable are described. Supportive figures, tables and graphics are included for each subsection (Figure 7).

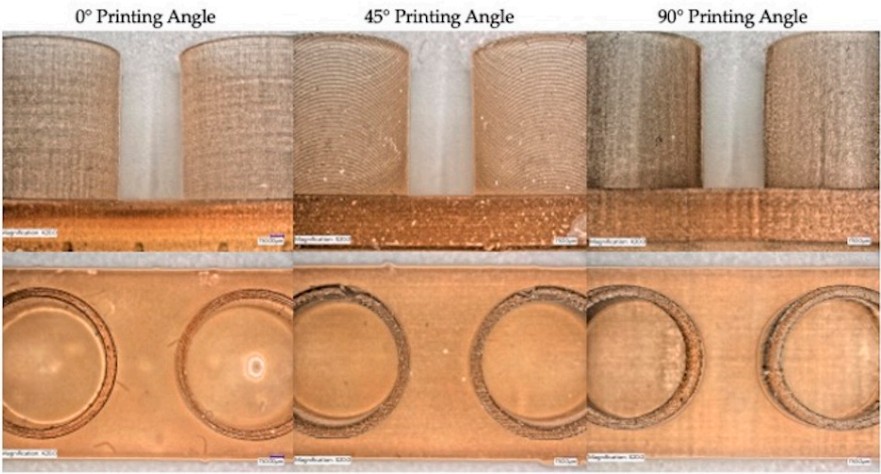

**Figure 7.** Representative photos of samples printed with different printing orientations showing discrepancies in distances and angle between cylinders in all the groups.

### 3.1. Angle between Cylinders

None of the 3D printing orientations resulted in perfect parallelism between cylinders. The 90 degrees printing orientation group had the closest angle to 0 degrees (best for parallelism) ($0.4143 \pm 0.2435$ degrees) $p < 0.001$ compared to the other printing orientation, 45 degrees ($1.4571 \pm 0.2378$ degrees) $p < 0.001$ and 0 degrees (worst for parallelism) ($2.7429 \pm 0.1886$ degrees) $p < 0.001$ (Figure 8, Table 1). The statistical comparison showed significant differences between the three groups (Table 2).

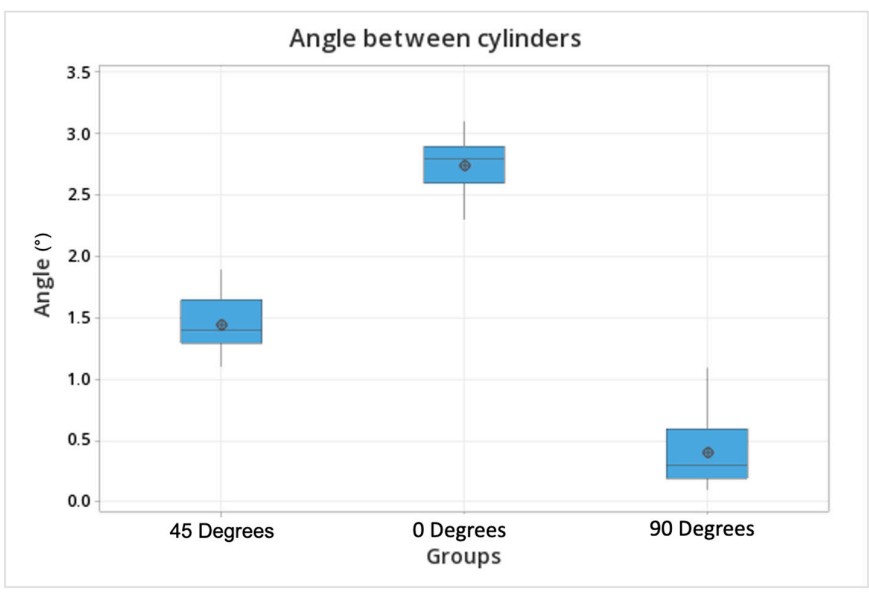

**Figure 8.** Box plot comparisons of the angle formed between cylinders printed at different angulations (45, 0, 90 degrees). The dots represent the mean values, the upper and lower lines are the upper and lower values within each group.

**Table 1.** Descriptive statistics of the angle between cylinders printed with different orientations. Values near 0 represent parallel cylinders.

| Angle | N | Mean | SD | 95% CI |
|---|---|---|---|---|
| CA 45 Degrees | 21 | 1.4571 | 0.2378 | (1.3591, 1.5552) |
| CA 0 Degrees | 21 | 2.7429 | 0.1886 | (2.6448, 2.8409) |
| CA 90 Degrees | 21 | 0.4143 | 0.2435 | (0.3162, 0.5124) |

**Table 2.** Statistical comparison of the angle formed between cylinders printed with different orientations. Tukey test.

| Difference of Levels | Difference of Means | SE of Difference | 95% CI | T-Value | Adjusted *p*-Value |
|---|---|---|---|---|---|
| CA (Angle) 0 Degrees–CA (Angle) 45 Degrees | 1.2857 | 0.0693 | (1.1190, 1.4524) | 18.54 | 0.001 |
| CA (Angle) 90 Degrees–CA (Angle) 45 Degrees | −1.0429 | 0.0693 | (−1.2095, −0.876) | −15.04 | 0.001 |
| CA (Angle) 90 Degrees–CA (Angle) 0 Degrees | −2.3286 | 0.0693 | (−2.4953, −2.161) | −33.58 | 0.001 |

### 3.2. Distance between Cylinders (Lateral View Top)

None of the groups had the distance of 4 mm between cylinders established in the reference STL model. The top of the cylinders converged to the center and the distances were shorter. The 45-degree printing orientation was closest to the reference of 4 mm (3733.05 ± 40.82 µm) compared to the other printing orientations (Figure 9, Table 3). The smallest distance was observed at 90-degree printing orientation (3537.53 ± 38.25 µm). The statistical comparison demonstrated significant differences between the means between all the groups (Table 4).

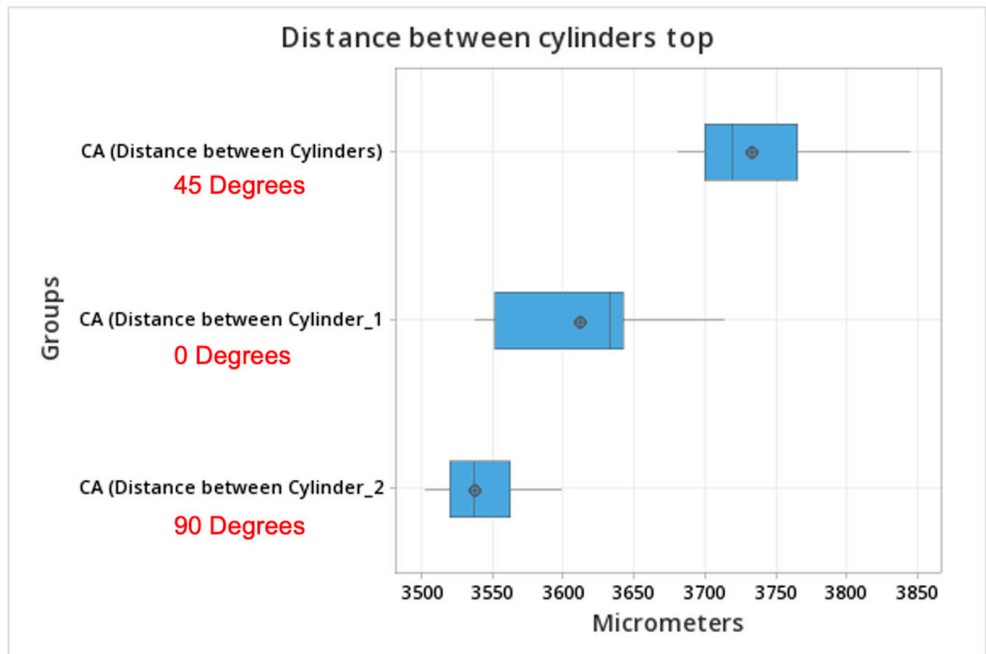

**Figure 9.** Box plot of the distance between the top of the cylinders printed with different orientations.

**Table 3.** Descriptive statistics of the distance between cylinders at the top.

| Factor | N | Mean | SD | 95% CI |
|---|---|---|---|---|
| CA DC 45 Degrees | 21 | 3733.05 | 40.82 | (3714.20, 3751.90) |
| CA DC 0 Degrees | 21 | 3611.3 | 49.6 | (3592.5, 3630.2) |
| CA DC 90 Degrees | 21 | 3537.53 | 38.25 | (3518.69, 3556.38) |

### 3.3. Distance between Cylinders (Lateral View Base)

None of the distances recorded at the base of the cylinder maintained the 4 mm distance of the STL reference model. The cylinders converged towards the center and the distances were shorter. The 90-degrees printing orientation group was closest to the reference of 4 mm (3873.2 ± 110.5 µm) compared to the other printing orientations (Figure 10, Table 5). The smallest distance was observed at 0 degrees printing orientation (3548.363 ± 39.39 µm). Group comparisons showed that the distances between cylinders at the base were different among all groups (Table 6).

**Table 4.** Group comparisons of the distances at the top of the cylinders. Tukey test.

| Difference of Levels | Difference of Means | SE of Difference | 95% CI | T-Value | Adjusted *p*-Value |
|---|---|---|---|---|---|
| CA DC 0 Degrees–CA DC 45 Degrees | −121.7 | 13.3 | (−153.8, −89.7) | −9.13 | 0.001 |
| CA DC 90 Degrees–CA DC 45 Degrees | −195.5 | 13.3 | (−227.6, −163.5) | −14.67 | 0.001 |
| CA DC 90 Degrees–CA DC 0 Degrees | −73.8 | 13.3 | (−105.8, −41.8) | −5.54 | 0.001 |

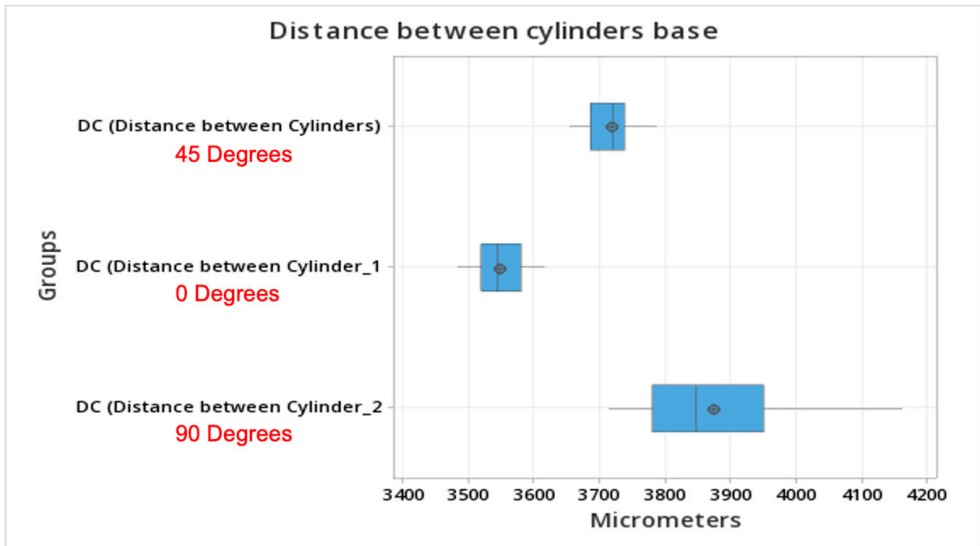

**Figure 10.** Distance at the base of cylinders printed with different orientations. Printing at a 90-degree angle resulted in values closer to the reference model. Meanwhile, orientations of 45 or 0 degrees resulted in shorter distances at the cylinders' bases.

**Table 5.** Descriptive statistics of the distance between cylinders at the base.

| Factor | N | Mean | SD | 95% CI |
|---|---|---|---|---|
| DC 45 Degrees | 21 | 3718.27 | 36.22 | (3687.33, 3749.21) |
| DC 0 Degrees | 21 | 3548.36 | 39.39 | (3517.42, 3579.30) |
| DC 90 Degrees | 21 | 3873.2 | 110.5 | (3842.3, 3904.2) |

**Table 6.** Group comparisons of the distance between cylinders at the base. Tukey test.

| Difference of Levels | Difference of Means | SE of Difference | 95% CI | T-Value | Adjusted *p*-Value |
|---|---|---|---|---|---|
| CA DC 0 Degrees–CA DC 45 Degrees | −121.7 | 13.3 | (−153.8, −89.7) | −9.13 | 0.001 |
| CA DC 90 Degrees–CA DC 45 Degrees | −195.5 | 13.3 | (−227.6, −163.5) | −14.67 | 0.001 |
| CA DC 90 Degrees–CA DC 0 Degrees | −73.8 | 13.3 | (−105.8, −41.8) | −5.54 | 0.001 |

### 3.4. Thickness of Cylinder Wall

Printing at 0 degrees provided the most values close to the 600 microns of the standard model. Printing at 45 degrees orientation produced a slight increment of the wall thickness (Figure 11). The 90-degrees group experienced the most discrepancy in the thickness of the wall of the cylinders (Figure 11). Printing at 90 degrees orientation showed thicker walls exceeding the standard by 400–500 microns (Figure 11).

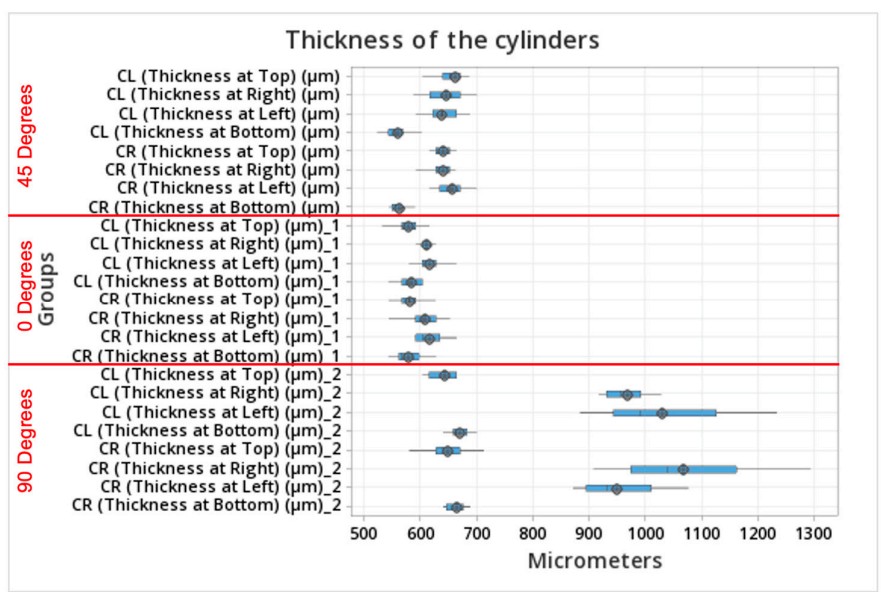

**Figure 11.** Box plot comparisons of the thickness of the cylinder walls at different angulations (45, 0, 90 degrees) measured from a coronal view.

*3.5. Global Evaluations*

3.5.1. Root Mean Square Error of the Cylinders (RMSE)

The root mean square differs slightly between all groups. The 90-degrees group had the lowest RMS error values (0.6158 ± 0.0511 mm) compared to the other printing orientations (Figure 12, Table 7).

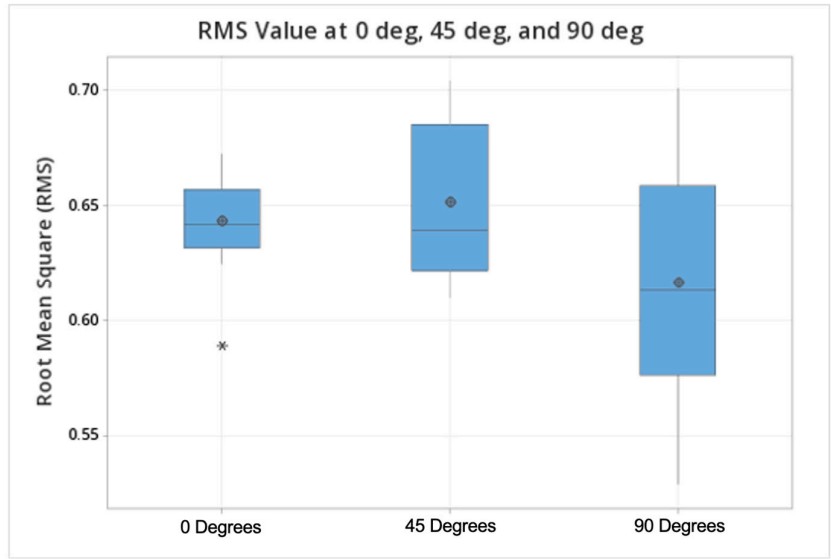

**Figure 12.** Box plot comparisons of the RMSE values at all orientations (0, 45, 90 degrees).

**Table 7.** Descriptive statistics of the RMSE values at all orientations (0, 45, 90 degrees).

| Sample | N | Mean | SD | SE Mean |
|---|---|---|---|---|
| 0 Degrees | 21 | 0.64334 | 0.01873 | 0.00409 |
| 45 Degrees | 21 | 0.65199 | 0.03332 | 0.00727 |
| 90 Degrees | 21 | 0.6168 | 0.0511 | 0.0112 |

The statistical comparison showed differences between the means between the 0-degrees group and 90-degrees group ($p = 0.015$), as well as between the 45-degrees group

and 90-degrees group ($p$ = 0.022). (Table 8). No difference was observed for the RMS values among the 0-degrees group and 45-degrees group ($p$ = 0.392).

**Table 8.** T-Value and $p$-Value comparisons of RMSE at all orientations.

| Samples | T-Value | $p$-Value |
|---|---|---|
| 0 Degrees vs. 45 Degrees | −0.88 | 0.392 |
| 0 Degrees vs. 90 Degrees | 2.67 | 0.015 |
| 45 Degrees vs. 90 Degrees | 2.48 | 0.022 |

### 3.5.2. Negative Mean Deviations

The negative mean deviations differ slightly between the three groups. The 90-degrees group had the lowest negative average deviations (−0.5747 ± 0.0665 mm) compared to the other printing orientations (Figure 13, Table 9).

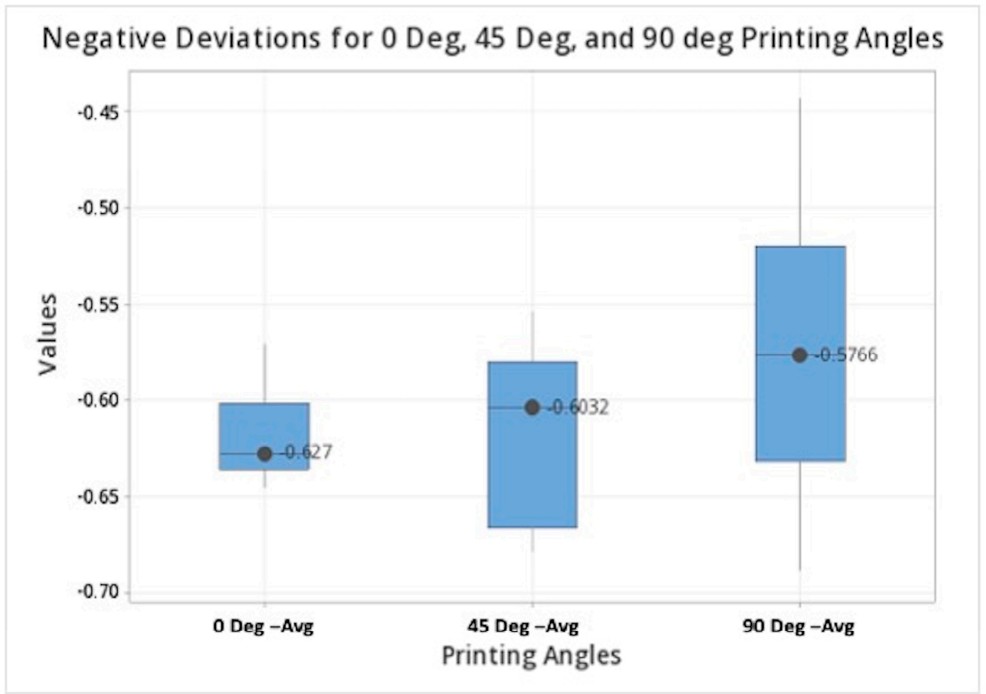

**Figure 13.** Box plot comparisons of the negative deviation values at all orientations (0, 45, 90 degrees).

**Table 9.** Descriptive statistics of the negative deviation values at all orientations (0, 45, 90 degrees).

| Variable | N | Mean | SE Mean | StDev | Minimum | Q1 | Median | Q3 | Maximum |
|---|---|---|---|---|---|---|---|---|---|
| 0 Degrees–Avg. | 21 | −0.61731 | 0.00475 | 0.02179 | −0.64520 | −0.63535 | −0.62700 | −0.60090 | −0.57010 |
| 45 Degrees–Avg. | 21 | −0.61815 | 0.00964 | 0.04417 | −0.67830 | −0.66545 | −0.60320 | −0.57950 | −0.55360 |
| 90 Degrees–Avg. | 21 | −0.5747 | 0.0145 | 0.0665 | −0.6885 | −0.6315 | −0.5766 | −0.5191 | −0.4427 |

The statistical comparison showed differences between the means between the 0-degrees group and 90-degrees group ($p$ = 0.006) and between the 45-degrees group and 90-degrees group ($p$ = 0.035) (Table 10). No difference was observed for the negative average deviations among the 0-degrees group and 45-degrees group ($p$ = 0.947) (Table 10).

### 3.5.3. Positive Mean Deviations

The positive mean deviations differed greatly between all groups. The 90-degrees group had the lowest positive average deviations (0.3795 ± 0.2176 mm) compared to the other printing orientations (Figure 14, Table 11). The statistical comparison showed significant differences between the means between all the groups (Table 12).

**Table 10.** T-Value and *p*-Value comparisons of the negative deviation values at all orientations.

| Samples | T-Value | *p*-Value |
|---|---|---|
| 0 Degrees vs. 45 Degrees | 0.07 | 0.947 |
| 0 Degrees vs. 90 Degrees | −3.05 | 0.006 |
| 45 Degrees vs. 90 Degrees | −2.26 | 0.035 |

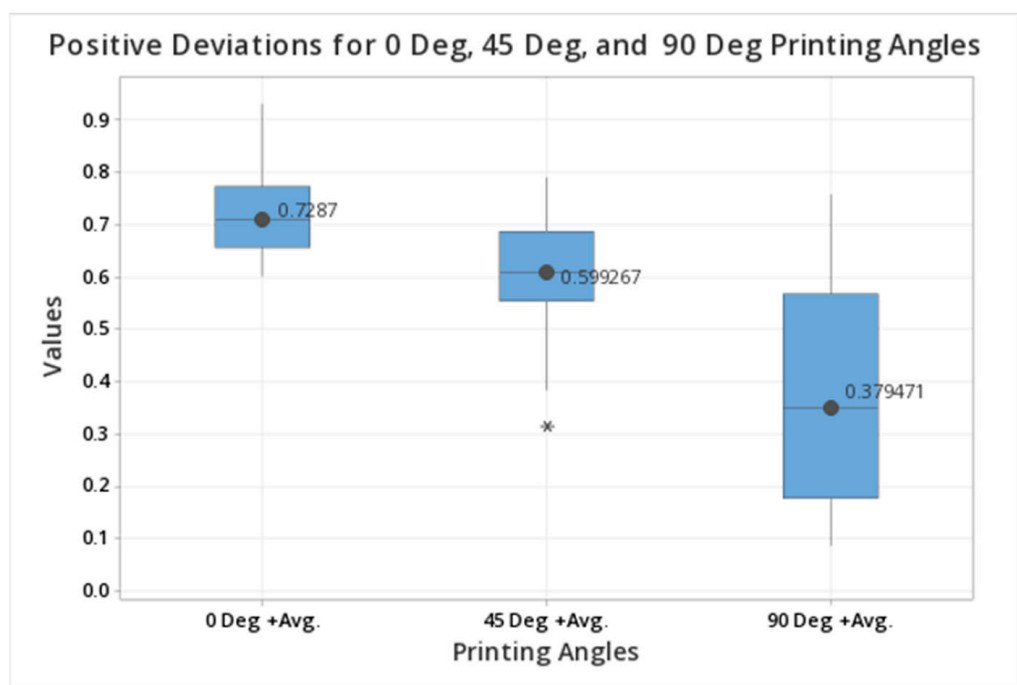

**Figure 14.** Box plot comparisons of the positive deviation values at all orientations (0, 45, 90 degrees).

**Table 11.** Descriptive statistics of the positive deviation values at all orientations (0, 45, 90 degrees).

| Variable | N | Mean | SE Mean | StDev | Minimum | Q1 | Median | Q3 | Maximum |
|---|---|---|---|---|---|---|---|---|---|
| 0 Degrees+ Avg. | 21 | 0.7287 | 0.0193 | 0.0885 | 0.6026 | 0.6573 | 0.7095 | 0.7737 | 0.9329 |
| 45 Degrees+ Avg. | 21 | 0.5993 | 0.0248 | 0.1135 | 0.3171 | 0.5551 | 0.6086 | 0.6877 | 0.7900 |
| 90 Degrees+ Avg. | 21 | 0.3795 | 0.0475 | 0.2176 | 0.0873 | 0.1788 | 0.3495 | 0.5685 | 0.7585 |

**Table 12.** T-Value and *p*-Value comparisons of the positive deviation values at all orientations.

| Samples | T-Value | *p*-Value |
|---|---|---|
| 0 Degrees vs. 45 Degrees | 4.17 | 0.001 |
| 0 Degrees vs. 90 Degrees | 6.53 | 0.001 |
| 45 Degrees vs. 90 Degrees | 5.12 | 0.001 |

3.5.4. Global Deviations at the Base Level

The bases presented similar root mean square errors (RMSEs) without differences among the different printing orientations. The statistical comparisons did not show differences between groups ($p > 0.05$). (Figure 15, Tables 13–15).

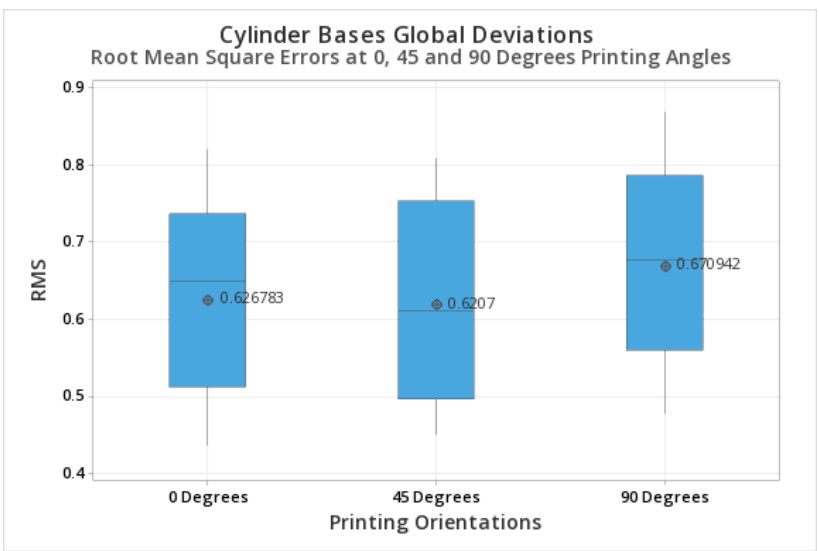

**Figure 15.** Box plot comparisons of the RMSE at the base of the cylinders at 0, 45, 90 degrees.

**Table 13.** Descriptive statistics of the RMSE values at all orientations at the bases (0, 45, 90 degrees).

| Factor | N | Mean | StDev | 95% CI |
|---|---|---|---|---|
| 0 Degrees | 21 | 0.6079 | 0.1278 | (0.5417, 0.6742) |
| 45 Degrees | 21 | 0.6090 | 0.1203 | (0.5428, 0.6753) |
| 90 Degrees | 21 | 0.6475 | 0.1194 | (0.5812, 0.7137) |

**Table 14.** ANOVA analysis for the RMSE at the base of the cylinders.

| Analysis of Variance | | | | | |
|---|---|---|---|---|---|
| Source | DF | Adj SS | Adj MS | F-Value | *p*-Value |
| Factor | 2 | 0.01419 | 0.007097 | 0.47 | 0.627 |
| Error | 39 | 0.58552 | 0.015013 | | |
| Total | 41 | 0.59972 | | | |

**Table 15.** Tukey comparisons for the RMSE at the base of the cylinders for all the printing orientations.

| Factor | N | *p*-Value |
|---|---|---|
| 90 Degrees | 21 | 0.6475 |
| 45 Degrees | 21 | 0.6090 |
| 0 Degrees | 21 | 0.6079 |

## 4. Discussion

The goal of the present study was to evaluate the influence of the printing angles on the parallelism, distance, and thickness of two adjacent cylinders on a simulated surgical guide. The results of this study showed that the printing angles significantly affected all these parameters.

The printing orientation and the orientation of the objects in the printing platform can influence the number of objects that can fit in the build platform, the number of supports, the accuracy, and the waste of material [32,34]. Specifically in implant surgical guides, the printing orientation affects the accuracy and level of distortion [32]. However, the isolated effects of the printing angle on the cylinders that guide the drilling and implant placement procedures has not been evaluated.

The results of the present study showed that the best parallelism between cylinders was achieved by printing at 90 degrees ($0.4143 \pm 0.2435$ degrees). Meanwhile, printing at 45 and 0 degrees produced deviations between 1.5 and 3 degrees, respectively. These results suggest that if the number of cylinders increases, then printing at 0 and 45 degrees can potentially increase the disparallelism between cylinders in a summative manner.

The distance between cylinders is measured from a lateral view (CA) and coronal view (DC). None of the distances recorded maintained the 4 mm distance of the STL reference model and the majority of distances recorded were less than 4 mm. The 45-degrees group ($3733.05 \pm 40.82$ µm) was closest to the reference when measured from a side view (CA) and the 90-degrees group ($3873.2 \pm 110.5$ µm) was closest to the reference when measured from the coronal view (DC). However, the 90-degrees group ($3537.53 \pm 38.25$ µm) produced the smallest distance when measured from the lateral view (CA), indicating a significant convergence of cylinders. The 45-degrees group recorded the most similar means when measured from the lateral (CA) view and top view (DC), indicating less convergence compared to the 0-degrees group and 90-degrees group. Therefore, the present results indicate that printing at 45 degrees will result in the most consistent distances between cylinders.

When analyzing the thickness of the cylinder walls, the 0-degrees and 45-degrees groups achieved values closer to the 600 microns of the STL reference model. The 90-degrees group had the greatest variation in the thickness of the wall in the cylinders, with the thickness at the right and left of both cylinders around 900 to 1000 microns. An increase in the thickness of the cylinder walls can influence the parallelism and the distance between cylinders.

The 90-degrees group had the lowest RMS values, negative mean deviations, positive mean deviations, and highest standard deviations compared to the other printing orientations. The global mesh comparisons indicate that the 90-degrees group had greater accuracy and fewer errors to the reference STL in comparison to the printing angles; however, all RMS values are comparable.

The surgical guide allows for the application of a virtually planned implant position into a clinical environment [35]. Any deviations between the planned and actual implant positioning are based upon the cumulative sum of errors during guide fabrication, post processing, intraoral or extraoral scans, data acquisition, and actual implant placement [35,36]. Based on the result of the present study, it can be observed that an increase in the number of cylinders may result in an increase in the cumulative errors between the initially planned position and the printed cylinders.

Multiple factors might lead to the seat distortion for 3D-printed surgical guides. For example, the curing process of each layer during 3D printing leads to the shrinkage of each layer during the polymerization process [37]. As each additional layer is cured, internal stresses and distortions may occur leading to a systematic deviation resulting in a guide smaller than intended [37]. In addition, further post-printing processes such as support struts, cleaning, and curing may potentially introduce an increase in dimensional errors and influence the seating of the guide [37,38].

In the present study, the printing orientation did not affect the global deviations of the cylinders' bases, which is probably related to the base thickness (3 mm) and the lack of complexity of the geometrical form of the base (rectangle). However, in conventional surgical guides prepared on teeth and considering the arch curvatures and other patient factors, the influence of the base design can't be overlooked. Additionally, the present report evaluated the accuracy of the 3D-printed bases. Future research with the same materials is needed to test mechanical properties such as flexural strength and hardness to achieve a complete overview of the materials tested [39,40]. Alternative materials with better optical and mechanical properties may produce better outcomes when manufacturing with different printing orientations.

Within the limitations of the present study should be noted the in-vitro nature of the experiment, that a single SLA printer was used, and that only three printing orientations

were tested. The strengths of this work lie in the following aspects: a simple geometric form with two parallel cylinders excluded the variability of a guide fabricated with the curvature of a dental arch; the calibration of the procedures resulted in reduced variability; in addition, the known dimensions of the reference model allowed a precise comparison with the multiple printed samples.

The practical and clinical implications of this work are that perfect surgical guides (equal to the design) do not exist, and surgical guides with multiple cylinders will have increased positional discrepancies produced by the sum of erroneous distances and angles between cylinders.

### 5. Conclusions

Within the limitations of this experimental in-vitro study it can be concluded that the printing orientation influences the angle, the distance, and the thickness between adjacent cylinders of a surgical guide. The printed objects differ from the reference model and the 90-degree orientation produces the highest variability in cylinder thickness, the best parallelism, and the best global concordance with the reference model.

**Author Contributions:** Conceptualization, R.D.-R. and H.B.; Formal analysis, R.D.-R. and H.B.; Investigation, A.A.; Methodology, R.D.-R. and H.B.; Resources, R.D.-R.; Software, A.A. and R.D.-R.; Supervision, R.D.-R.; Validation, A.A., R.D.-R. and H.B.; Visualization, R.D.-R.; Writing—original draft, A.A.; Writing—review & editing, A.A., R.D.-R. and H.B. All authors have read and agreed to the published version of the manuscript.

**Funding:** This research received no external funding.

**Institutional Review Board Statement:** Not applicable.

**Informed Consent Statement:** Not applicable.

**Data Availability Statement:** Data will be available upon request to the corresponding authors.

**Acknowledgments:** The authors acknowledgment the support of the center of Center for Implant Digital Technology (CIDT) and Axel Calderon for their technical support. The authors also thank the Digital Implant Prosthodontics Research Laboratory (DIPRESLAB) for its support and providing the microscope for this study.

**Conflicts of Interest:** The authors declare no conflict of interest.

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
