# Peer review of "Influence of the Printing Orientation on Parallelism, Distance, and Wall Thickness of Adjacent Cylinders of 3D-Printed Surgical Guides"

_prosthesis, doi:10.3390/prosthesis5010023_

Round 1

Reviewer 1 Report

This study was designed to evaluate the influence of the printing angles on the parallelism, distance, and thickness of two adjacent cylinders on a simulated surgical guide. The results showed that the printing angles significantly affected all these parameters and the 90-degree orientation produces the highest variability in cylinder thickness, the best parallelism, and the best global concordance with the reference model. It's an interesting study and does provide some new ideas to the readers. However, there're still one issue which should be addressed.

The topic of this manuscript is about 3D printing technology and has little to do with dental prostheses. In order to solve this problem, it is suggested that the author add the implant guide plate as the sample to conduct the experiment again, or add the in vivo experiment.

So, major revision should be recommended for this manuscript.

Author Response

Answers to Reviewer 1

Dear Reviewer 1 thanks for your time completing the review of our manuscript.

In regards to your comments

“This study was designed to evaluate the influence of the printing angles on the parallelism, distance, and thickness of two adjacent cylinders on a simulated surgical guide. The results showed that the printing angles significantly affected all these parameters and the 90-degree orientation produces the highest variability in cylinder thickness, the best parallelism, and the best global concordance with the reference model. It's an interesting study and does provide some new ideas to the readers.

However, there're still one issue which should be addressed.

The topic of this manuscript is about 3D printing technology and has little to do with dental prostheses. In order to solve this problem, it is suggested that the author add the implant guide plate as the sample to conduct the experiment again, or add the in vivo experiment.

So, major revision should be recommended for this manuscript.”

We respectfully disagree with reviewer-1 comment.  

This article was submitted to the journal Prosthesis’ special issue entitled “Oral implantology: Current aspects and Future perspectives”.  Among the research areas that this special edition covers are listed: Implantology and digital workflow.

Our article is relevant to implantology and digital workflow aspects. Also, our article is relevant to prosthodontics. The influence of the implant position on implant restoration is undeniable.

For example, misaligned implants result in complicated prosthetic techniques that require primary frameworks, unconventional bar designs, and esthetic complications. In addition, non-axial loads applied to the implants can increase mechanical failure. When implants are digitally planned for immediate loading, precise implant placement and fit of the CADCAM restorations is a pre-requisite. Therefore, reducing the errors of surgical guides favors correct implant positions and improves prosthetic treatment outcomes.

We have the data from the cylinder’s bases, but given that we wanted to determine how much the cylinders add to the deviations in surgical guides, that data was not added to this study. However, following your recommendations, we included information as supplementary material with the comparisons from the data from the cylinder’s bases.

Reviewer 2 Report

Authors submit an article that addresses impact influence of the printing orientation on parallelism, distance, and wall thickness of adjacent cylinders of 3D-printed surgical guides. 

Introduction

I don't recommend using "excellent repeatability, fast fabrication, sufficient accuracy", they do not have a quantitative statement. What a fast fabrication, compared to what? What is the ideal production time?

How often are surgical guides used in dental practice? How were they made before AM or subtractive manufacturing?

Materials and Methods

How many surgical guides could you print in one 3D print? Have you considered comparing placement on the betting board as well? Does location have an effect on production accuracy?

Figure 2. - highlight the building plate, and maybe I would complete the figure with a support structure.

Was the layer thickness of 100 microns given by the manufacturer, or did you have a choice of several options? If you had a choice, what made you decide on 100 micron layers.

lines 114 - fill in data on miktorskor - company, country

Results

-unify the formatting of subsection 

-lines 184 - repair "3d"

- lines 185-187 - complete units of degree 0° and 45°

- in graphs, add units on the y axis (angle [°], ....)

- figure 9, 10,11 - I recommend changing the description to the x-axis, an unnecessarily long name at the expense of the quality of the graph

I recommend unifying the units deg, degrees, ° throughout the article.

The article is written very simply, supported by appropriate literature. The obtained data are appropriately processed and the results presented in tabular and graphic form.

Author Response

Answers to Reviewer 2.

Dear Reviewer 2, sincere thanks for your time and effort in reading our manuscript.

According to your recommendations, the following changes and modifications were completed.

Introduction

I don't recommend using "excellent repeatability, fast fabrication, sufficient accuracy", they do not have a quantitative statement. What a fast fabrication, compared to what? What is the ideal production time?

Thanks for your comment.  The words “excellent, fast, and sufficient were removed and replaced by “repeatability and precision” which are supported by reference number 5.

How often are surgical guides used in dental practice? How were they made before AM or subtractive manufacturing?

Thanks for your comment. We added references to the previous methods of fabrication of surgical guides. In addition, we added references to current use of surgical guides and the reference numbers were adjusted within the manuscript and in the references section.

4.Umapathy T, Jayam1, Anila B, Ashwini C. Overview of surgical guides for implant therapy

J Dent Impl. 2015;5:48-52

  1. Chen P, Nikoyan M. Guided Implant Surgery A Technique Whose Time Has Come. Dent Clin NorthAm.2021;   65:67–80

Materials and Methods

How many surgical guides could you print in one 3D print?

Thanks for your question. That depends on the object that is being printed. In the case of our sample dimensions, we were able to print 21 samples in each batch.

Have you considered comparing placement on the betting board as well? Does location have an effect on production accuracy?

Many thanks for your comment. Yes, location may have an impact on the accuracy of the printed objects. Therefore, in the printing layout we distributed the objects away from the borders of the printing platform and evenly at the central area.  We also include this  in the discussion section.

Figure 2. - highlight the building plate, and maybe I would complete the figure with a support structure.

Thanks for your comment. We included figure 2b that illustrates the printing orientations and the  printing platforms.

Was the layer thickness of 100 microns given by the manufacturer, or did you have a choice of several options? If you had a choice, what made you decide on 100 micron layers.

Many thanks for your comment. The printer used in this experiment, layer thickness of 25, 50 or 100 microns. We decided a printing layer thickness of 100microns based on printing efficiency, and in our manuscript published in 2021. We observed that neither the dimensions,  storage time, or surface characteristics were affected by the printing layer thickness.

In addition, given that all our objects were printed with the same layer thickness, the impact of the layer thickness can be excluded as a cofounding factor.

Sabbah A, Romanos G, Delgado-Ruiz R. Impact of Layer Thickness and Storage Time on the Properties of 3D-Printed Dental Dies. Materials (Basel). 2021 Jan 21;14(3):509. doi: 10.3390/ma14030509.

lines 114 - fill in data on miktorskor - company, country

Thanks for your comment. The missing information was added.

Results

-unify the formatting of subsection 

Subsection was unfied

-lines 184 - repair "3d"

Corrected

- lines 185-187 - complete units of degree 0° and 45°

Completed

- in graphs, add units on the y axis (angle [°], ....)

Added

- figure 9, 10,11 - I recommend changing the description to the x-axis, an unnecessarily long name at the expense of the quality of the graph

I recommend unifying the units deg, degrees, ° throughout the article.

Done

The article is written very simply, supported by appropriate literature. The obtained data are appropriately processed and the results presented in tabular and graphic form.

Many thanks for your comments we found all relevant and correct. We believe that with the changes completed in the revised version of this manuscript, the quality and clarity are improved and now deserves publication.

Round 2

Reviewer 1 Report

The question raised has not been resolved. Therefore, rejection should be recommend.

Author Response

Dear Reviewer 1

In our first response, we explained that the study's goal was to evaluate the printing orientation on the distance between cylinders, their angulation, and wall thickness because the cylinder is the primary factor guiding the implant bed preparation and implant placement in fully guided implant surgery.

Reviewer 1 wanted to evaluate the bases. The bases do not have geometrical forms or landmarks that can be assessed beyond the cylinders, and these were considered in our study.  In addition, in this in-vitro model, we excluded arch curvatures, the presence of teeth, mouth opening, and other confounding factors. Thus, by isolating the cylinders, the precision of the study increases.

However, trying to clarify the reviewer 1 concerns, we are providing now  the data from the cloud comparisons from the printed bases and the statistical analysis and comparisons from those in the material and methods and results section.  The statistical comparisons did not show differences between the root mean square errors at the base level and the different printing orientations, supporting the relevance of studying the cylinders not the bases. 

Another highlighted aspect  is that the methodology used in this study is innovative, and Reviewer 2 did not find problems with the methods used in the study.

We hope that the additional information clarifies the reviewer 1 concerns.

Respectfully, 

The authors

Sincerely,

The authors.

Round 3

Reviewer 1 Report

I accept the authors explanation. The manuscript could be accepted for the publication.

Author Response

Many thanks for your time and expertise during the review of our manuscript. 

Sincerely, The authors.